

# Anthropocene geochemistry of metals in sediment cores from the Laurentian Great Lakes

Malachi Nicholas Granmo, Euan D. Reavie, Sara P. Post and Lawrence M. Zanko

Natural Resources Research Institute, University of Minnesota Duluth, Duluth, MN,
United States of America

## ABSTRACT

Geochemical analyses applied to lake sedimentary records can reveal the history of pollution by metals and the effects of remedial efforts. Lakes provide ideal environments for geochemical studies because they have steady deposition of fine grained material suitable for fixation of pollutants. The Laurentian Great Lakes are the most studied system in this field, and they have well-preserved chronological profiles. To date, this important system has been considered in parts for inorganic geochemistry, hampering basin-wide conclusions regarding metal contamination. We filled spatial and temporal gaps in a comprehensive geochemical analysis of 11 sediment cores collected from all five Great Lakes. Hierarchical cluster analysis of all Great Lakes samples divided the metal analytes into five functional groups: (1) carbonate elements; (2) metals and oxides with diverse natural sources, including a subgroup of analytes known to be anthropogenically enriched (Cd, Pb, Sn, Zn, and Sb); (3) common crustal elements; (4) metals related to coal and nuclear power generation; and (5) all of the co-occurring rare earth elements. Two contamination indices ($I_{geo}$ and EF) applied to sedimentary metals indicated that Na, Co, Mn, Cd, Pb, Ta, and Cu were each, at some point during the Anthropocene, the most enriched metal pollutants in Great Lakes sediments. Land uses correlated with the metal analytes, such as increases in contaminant metals with the rise in catchment population and increases in carbonate elements (e.g. Ca) with agriculture. Certain contamination trends were observed basin-wide, such as for the atmospheric pollutant Pb, which followed a rise associated with fossil fuel combustion and a decline following the ban of leaded gasoline. Other trends were lake-specific, such as recent high concentrations of Na in Lake Superior, likely due to road salt applications, and a late-20th-century peak in Ca associated with algal whiting events in Lake Ontario. Some metals exceeded guidelines for sediment quality, in some cases prior to European settlement of the basin, indicating that a paleolimnological context is important for appropriate management of sediment contamination. The Great Lakes are sensitive to environmental changes such as pollution by metals, and it is clear that while there has been remedial success, results from the uppermost intervals of cores indicate ongoing problems.

Corresponding author
Malachi Nicholas Granmo,
aliff002@d.umn.edu

# INTRODUCTION

The Laurentian Great Lakes have a complex history of contamination. While Native American influence on the environment of the Great Lakes was likely minimal and localized, such as copper mining effects around Isle Royale, Lake Superior (*Pompeani et al., 2015*), European westward expansion was marked by overfishing, logging, mining, and the introduction of non-native species. Increasing environmental concentrations of metals in the region in the 1850s correspond with extensive European settlement and with the growth of population and industry, increasing even more rapidly due to the burning of fossil fuels. Contamination of the lakes increased until pollution reached such a critical level that the governments of the United States and Canada felt compelled to act (*Sly & Munawar, 1988*; *Beeton, 2002*) with the passing of the Clean Water Act (*United States of America, 1972*) and the Great Lakes Water Quality Agreement (*Canada & United States of America, 1972*).

In order to assess the historical and continuing environmental impacts of contamination and the efficacy of cleanup efforts, a long-term account of environmental metals is needed. Such assessments have been supported by paleolimnological investigations. Metals are useful for assessing historical pollution in lakes, as they are among the best characterized contaminants for monitoring. Pollutants are mobilized in rivers, move away from their sources, and are incorporated into lacustrine sediments. Several pollutants are also deposited via the atmosphere. The pollutants are preserved due to affinity with sediments, the fact that they do not degrade, and most of them do not mobilize (though there are some exceptions such as aqueous forms of Fe and Mn). This kind of geochemical analysis is useful in systems where the sediment record is minimally disturbed by bioturbation or flow (*Alderton, 1985*). Heavy metals are among the best characterized contaminants for environmental monitoring (especially Hg and Pb), as they are persistent in the sediments (*Heim & Schwarzbauer, 2012*). However, a comprehensive study of metals and other inorganic chemicals throughout the entire Great Lakes basin has not been done. Studies to date have been generally limited to a few inorganic contaminants or geographically limited portions of the Great Lakes basin. This report is the first comprehensive geochemical analysis of metals for the whole Great Lakes basin, with a major goal being historical reconstruction of natural conditions, anthropogenic impacts, and remediation.

Many paleolimnological studies in the Great Lakes have focused exclusively on the highly toxic element Hg: *Kennedy, Ruch & Shimp (1971)*, *Pirrone et al. (1998)*, *Rossmann (2010)*, and *Yin et al. (2016)* in Lake Michigan; *Kovacic (1972)*, *Walters et al. (1972)*, *Walters, Wolery & Myser (1974)*, *Wolery & Walters (1974)*, *Pirrone et al. (1998)* in Lake Erie; and *Thomas (1972)*, *Breteler et al. (1984)*, *Pirrone et al. (1998)*, *Marvin et al. (2004)* in Lake Ontario. It was generally observed that Hg levels in sediment records increased beginning in the mid-1800s, reached a maximum in the 1970s, and decreased with recent industrial regulations. Several studies have also exclusively tracked historical Pb concentrations as a tracer of leaded gasoline combustion using sediment cores in the Great Lakes: *Edgington, Robbins & Karttunen (1974)*, *Robbins & Edgington (1974)*, and *Edgington & Robbins (1976)* in Lake Michigan; *Ritson et al. (1994)* in Lake Erie, *Rossmann, Pfeiffer & Filkins (2014)* in

Lake Michigan; *Rossmann, Pfeiffer & Filkins (2014)* in Lake Huron; and *Graney et al. (1995)* throughout the entire Great Lakes Basin. Overarching findings were that Pb concentrations increased in the 20th century but more recently declined due to the legislated transition to Pb-free gasoline.

In general, Great Lakes geochemical studies in sediment cores have focused on individual lakes. *Nussmann (1965)* chose Lake Superior as the first lake for this kind of study because it was viewed as the least anthropogenically impacted lake and thereby represented a baseline for comparison with other lakes. Subsequent studies included Lake Michigan (*Shimp, Leland & White, 1970*), Lake Ontario (*Cronan & Thomas, 1972*), Lake Erie (*Walters, Wolery & Myser, 1974*), and eventually Lake Huron (*Robbins, 1980*). A full review of this previous work is presented by *Aliff et al. (2020)*.

We aimed to fill spatial and temporal gaps and to cover a wide range of metals analytes in a comprehensive geochemical analyses of 11 sediment cores from the five Great Lakes. Because of likely lake-specific interests to Great Lakes researchers, we spend time on descriptive interpretations relative to each lake. We further aimed to relate geochemical history to past anthropogenic activities, both detrimental and remedial. This included stratigraphic surrogates for natural deposition due to erosion of soils, and bedrock and human activities such as mining, tailings disposal, and burning of fossil fuels. In performing a basin-wide investigation, we anticipated similarities in widespread stressors such as atmospheric lead deposition (*Norton et al., 1990*) but that lake-specific trends would correspond with each lake's unique anthropogenic history and physico-chemical characteristics.

## METHODS

### Coring

Eleven sediment cores were collected per *Reavie et al.* (*2017*; Fig. 1). Three cores were collected from Lake Superior, and two cores were collected from Lakes Huron, Erie, Ontario, and Michigan. Cores were collected from the USEPA's R/V Lake Guardian or the University of Minnesota's R/V Blue Heron using an Ocean Instruments model 750 box corer, from which two 6.5-cm internal diameter cylindrical cores were subsampled, or an Ocean Instruments model MC-400 multi-corer (9.4 cm diameter cores). Temporal records ranged from ~100 y to over 300 y prior to the coring date.

### Extrusion and dating

For each location, one core was extruded at fine intervals (as fine as 0.25 cm in upper intervals to 1 cm intervals at the bottom of the core), depending on estimated accumulation rates and need for temporal resolution. Isotopic dating to develop temporal records for each core follows methods described by *Shaw Chraïbi et al. (2014)*. All cores had exponential $^{210}$Pb profiles indicating typical isotopic decay with time, and errors associated with dates ranged from 1–2 yr in the most recent three decades to +/- 10–20 yr around ca. 1850. Because background concentrations of $^{210}$Pb were not achieved, the western Lake Erie core dating included supplementary $^{137}$Cs analysis to pinpoint the 1963 peak resulting from

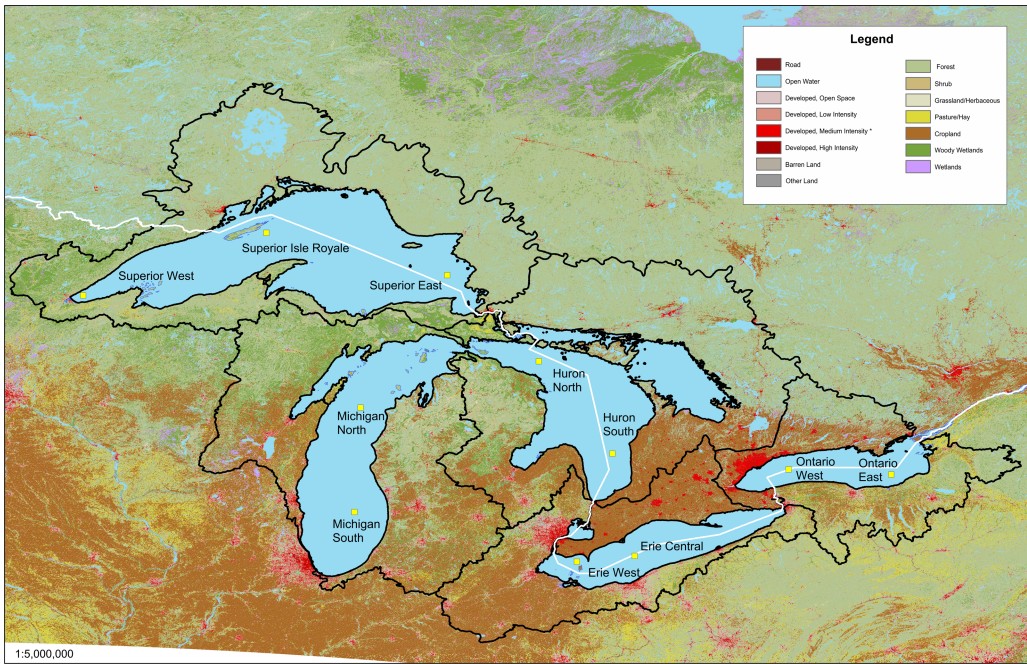

**Figure 1** **Land use map of the Great Lakes basin watershed.** Each lake's catchment is depicted and the 11 sediment core locations are shown. United States land use data is taken from the National Land Cover Database and Canadian land use data is taken from the open.canada.ca website. The classifications for each country were simplified and combined. The Canadian data does not separate development by intensity so all of the Canadian developed land is coded as medium intensity.

weapons testing (*Appleby, 2002*). Complete dating models for these cores are provided in the Supplementary Materials.

## Historical stressors

Spatial and temporal land use patterns around the Great Lakes were collected from historical records going back as far as the late 1700s (*Reavie, Cai & Brown, 2018*). Land uses included were agriculture, population, mining, and forest cover. The level of each land use was summarized within watersheds covering the Great Lakes basin. We selected sub-watersheds around each lake to summarize historical lake-specific stress. These data and full details on their sources are available in a public archive provided by *Reavie, Cai & Brown (2018)*.

## Chemical analyses

Sedimentary analyses for metals and oxides were performed by personnel at the University of Minnesota, Department of Earth Sciences, Analytical Geochemistry Laboratory. Excepting modern instruments, these methods largely follow (*Malo, 1977*) for extraction and analysis of metal analytes. For equipment calibration, standard reference materials and sample blanks were assessed every time a new core was being analyzed. For trace metals (As, Ba, Cd, Ce, Co, Cr, Cs, Cu, Dy, Er, Eu, Ga, Gd, Hf, Ho, La, Lu, Mo, Nb, Nd, Ni, Pb, Pr, Rb, Sb, Sc, Sm, Sn, Sr, Ta, Tb, Th, Tm, U, V, Y, Yb, Zn, Zr) analysis,

sediment subsamples were freeze-dried, and $0.25 \pm 0.02$ g of dry sediment were added to a 50-mL centrifuge tube. To this, 25 mL 0.5 N HCl was added and samples were heated at 80–85 °C in a hot-water bath for 30 min. Vials were transferred to an ice-water bath and allowed to cool for 5 min. Samples were centrifuged at 2,000 rpm for 10 min and then 10.0 mL of the supernatant was moved to 125-mL acid-washed poly-bottles. Each sample was diluted with $40 \pm 0.5$ g deionized water. Samples were assessed using inductively coupled plasma mass spectrometry (ICP-MS), which is capable of the determination of a range of metals and several non-metals (*B'Hymer, Day & Caruso, 2000*; *Jarvis & Jarvis, 1992*). Instrumentation was a Thermo Scientific XSERIES 2 ICP-MS (inductively coupled plasma—mass spectrometer) with ESI PC3 Peltier cooled spray chamber, SC-FAST injection loop and SC-4 autosampler. Elements were analyzed using He/$H_2$ collision-reaction mode. For quality control, all samples were analyzed three times and flagged if any single analysis of a given analyte varied by 10% from any other analysis for that analyte. No such discrepancies were noted so analyses as presented are averages of the three analyses.

For major metals (Al, Ca, Fe, K, Li, Mg, Mn, Na, P, Si, Ti), HCl-extracted samples were analyzed using a Mira Mist Peek nebulizer paired with a Thermo Scientific iCAP 6500 dual view ICP-OES (inductively coupled plasma—optical emission spectrometer) with the following settings: power 1150 W, nebulizer flow 0.65 L/min, cooling gas 12 L/min, auxiliary gas 0.5 L/min, 1 mL/min sample flow rate, 8 sec/replicate and 5 replicates/sample. For analysis of oxides, freeze-dried sediment subsamples ($\sim$0.5 g) were processed directly using the same settings as those for major metals. Due to its low detection limit requirement and high cost of analysis, geochemical data for mercury (Hg) were not generated. For quality control, all samples were analyzed twice and flagged if the analyses for a given analyte varied by more than 10%. No such discrepancies were noted so analyses as presented are averages of the two analyses.

Here, we present geochemical data as concentrations relative to sediment dry weight (ppm). Not presenting accumulation rate data was partly for the sake of brevity, but we noted that profiles of metal accumulation rates were similar to those for concentrations, so we do not anticipate that the presentation of accumulation rates would have a notable effect on major trends. Further, as detailed by *Engstrom & Wright (1984)*, there are several reason to preferentially choose concentrations. Most notably in our case, several metals (such as Fe) are highly mobile in Great Lakes sediments, so calculated accumulation rates would falsely represent the actual accumulation at the time of deposition. Despite caution on the use of these data, we present our calculated accumulation rates as Supplementary Materials.

## Mapping

The land use map (Fig. 1) was compiled using ArcGIS 10.4. The map has a spatial resolution of 30 m. The US Land Use data was accessed through the mrlc.gov, the Multi-Resolution Land Characteristics Consortium. The data are part of the National Land Cover Database and the raster file was downloaded on May 14, 2019. The Canadian Land Use data were accessed through the open.canada.ca website, a project of Open Canada. The maps were produced by Agriculture and Agrifood Canada and TIFF files were downloaded on May

14, 2019. Classifications for both countries were combined which involved simplifying and combining land-cover classifications for each country.

## Data analysis

The following analyses were performed in R (*R Core Team, 2018*): cluster analysis, principal components analysis (PCA) and plotting of analyte temporal profiles. All analyses were done entirely in base R except for PCA which was plotted with the package maptools in order to minimize overlap of the analyte labels (*Bivand & Lewin-Koh, 2017*).

Hierarchical cluster analysis of the Great Lakes geochemical dataset focused on determining relationships among analytes (metallic elements and oxides). The cluster analysis used an unweighted pair group method with arithmetic mean by taking Euclidean distances of the correlation matrix. Ultimately, clustering was used to identify groups of analytes that exhibited similar trends in the last 150–200 years across all Great Lakes cores. A preliminary detrended correspondence analysis (DCA) indicated a first-axis gradient less than one standard deviation unit (i.e., low overall environmental variability), so PCA (a linear method) was chosen over other ordination methods. PCA using singular value decomposition of the centered and scaled data matrix allowed us to further illustrate relationships among these groups and to trace the temporal trajectories of each sediment core over time relative to the analyte scores. Analyte concentrations were scaled in order to avoid placing emphasis on analytes with naturally higher values.

Two indices were applied to determine the extent and threat of anthropogenic contamination in the sediment records: the geoaccumulation index ($I_{geo}$) and enrichment factor (EF). $I_{geo}$ is a measure used to compare present day contaminant concentrations with pre-impact background levels in order to better quantify anthropogenic impacts (*Stoffers et al., 1986*). For a select set of contaminant elements, $I_{geo}$ was calculated according to the following formula from *Müller (1969)*:

$$I_{geo} = log_2(C_n/1.5B_n) \tag{1}$$

where $C_n$ is the concentration of a given element $n$ in ppm in a sediment interval that is considered anthropogenically influenced (above the baseline intervals) and $B_n$ is the average concentration in ppm of the element in the pre-impact intervals of the sediment core. The 1.5 multiplication factor was introduced by *Stoffers et al. (1986)* to include variation in background values, which could be attributed to natural variations in sediment properties. Pre-impact (background) intervals for each core were selected based on plots of historical stressors (*Reavie, Cai & Brown, 2018*; Fig. 2). A lake was considered impacted based on a horizon of increased population, increased agricultural acreage, or increase in mining activity.

EF is a measure of sediment contamination relative to baseline (i.e., pre-impact) conditions adjusted for sediment composition. It minimizes contaminant variability associated with clay-rich mud/sand ratios by normalizing the contaminant content with respect to a reference element that acts as a proxy for the clay content of the sediment (*Abrahim & Parker, 2007*). EFs were calculated according to the following equation adapted

 

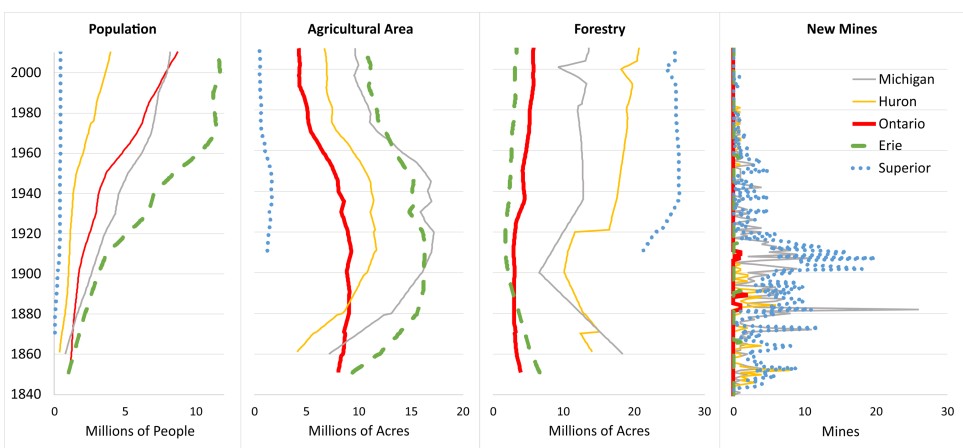

**Figure 2** **Plots of historical watershed stressors in the Great Lakes.** Data are compiled from *Reavie, Cai & Brown (2018)*.

from *Weiss et al. (1999)*:

$$EF = \frac{\left(\frac{Contaminant\ ppm}{Al\ ppm}\right)_{sediment}}{\left(\frac{Contaminant\ ppm}{Al\ ppm}\right)_{background}} \tag{2}$$

Where *sediment* is the sediment layer in the post-impact interval of interest and *background* is the sediment interval considered pre-impact. The Al concentration is used in this case as a reference element, as it is conservative and a major constituent of clay minerals. Aluminum has previously been used as a reference element in the Great Lakes by *Kemp & Thomas (1976)* and *Kemp et al. (1976)*.

Within each core, $I_{geo}$ and EF values were averaged within each decade following the temporal horizon representing the "impact" period. Lake Erie's western core was excluded from these analyses because the core only went back to 1930, which was after the start date of significant human impacts. Sodium concentrations were not included for Lake Huron South or Lake Michigan South as initial Na values were zero in both of these cores, hence there was no way to quantify enrichment relative to baseline.

Historical stressors of agriculture area, forested area, mining stress, new mining sites, overall and urban population were summarized for 60 watersheds of the Laurentian Great Lakes basin from early 1800 to 2010 (*Reavie, Cai & Brown, 2018*). The relationships between metal concentrations and historical stressors of drainage watersheds were analyzed by checking Pearson's correlation. The annual data of historical stressors were summarized as the sum, area weighted average and lake water volume weighted data, then linearly interpolated to sediment interval dates. Data from the 11 sediment cores were combined for analysis against stressors. Pearson correlation analyses were performed in R 3.5.1 using the function rcorr in the Hmisc package (*Harrell, 2019*).

## Contamination and toxicity guidelines

Sediment contamination guidelines help managers assess risk to biota and other water uses. In the Great Lakes region, Ontario set guidelines using the screening level concentration (SLC) approach. The SLC approach uses field data to examine the relationships between benthic organisms and contaminant levels. The SLC represents the concentration that 95% of species can tolerate. For metal contaminants, the Ontario guidelines have two levels: the lowest effect level, which indicates sediment that is clean to moderately polluted, and the severe effect level, which indicates heavy pollution that is likely to negatively impact the health of benthic organisms (*Persaud, Jaagumagi & Hayton, 1993*). Another set of contamination guidelines in use are the Effects Low Range (ER-L) and the Effect Medium Range (ER-M) developed by NOAA in marine systems. ER-L values represent the lower 10th percentile for biological impact and the median was represented by the ER-M. These guidelines are not used as official standards or criteria but can be used to help managerial decision making (*Long & Morgan, 1990*; *Long et al., 1995*). Yet another approach for calculating sediment guidelines is the apparent effects threshold (AET), which reports the sediment concentration of a contaminant above which significant biological effects always occur in benthic organisms. Apparent effects threshold-low (AET-L) and apparent effects threshold-high (AET-H) values were based on data from Puget Sound (*Barrick et al., 1988*). For our purposes, we used the Ontario guidelines when available, followed by the ER-L and ER-M values, and we used AET values when neither of the other sets of guidelines was available for a given metal contaminant. Even with all three of these guidelines, only a small subset of the metals had relevant guidelines.

# RESULTS

## Cluster analysis

The dendrogram formed by hierarchical cluster analysis of all Great Lakes samples (Fig. 3) divided the analytes into five functional groups.

- Group 1 contains only Ca, Mg and their oxides. Calcium and Mg are common carbonate elements (*Callender, 1969*; *Kemp et al., 1976*) that come from terrigenous clay minerals and they are also associated with carbon cycling in lakes (*Sgro & Reavie, 2017*). Microorganism respiration creates $CO_2$, which dissolves Ca from its carbonate minerals. Peaks can represent historical algae blooms and other spikes in primary productivity (*Yuan, 2017*).

- Group 2 contains analytes with diverse natural sources, though some are considered anthropogenic pollutants. For instance, sediment Na may be related to road salt input and $P_2O_5$ (phosphorus pentoxide) may come from fertilizer applications (*Chambers et al., 2016*). Many Group 2 elements are also known to be strongly associated with Fe and Mn oxides, which they sorb to strongly, especially Pb, Cu, Zn, Ni, Cd, and Co (*Stone & Marsalek, 1996*). Group 2 is further subdivided into Group 2a, a set of analytes known to be highly enriched in the Great Lakes (Cd, Pb, Sn, Zn, and Sb; *Robbins, 1980*; *Yuan, 2017*). Group 2a elements are also all atmospheric and specifically enriched due to smelting of Pb and Cu (*Crecel & Johnson, 1974*; *Warren, 1981*; *Galloway et al., 1982*;

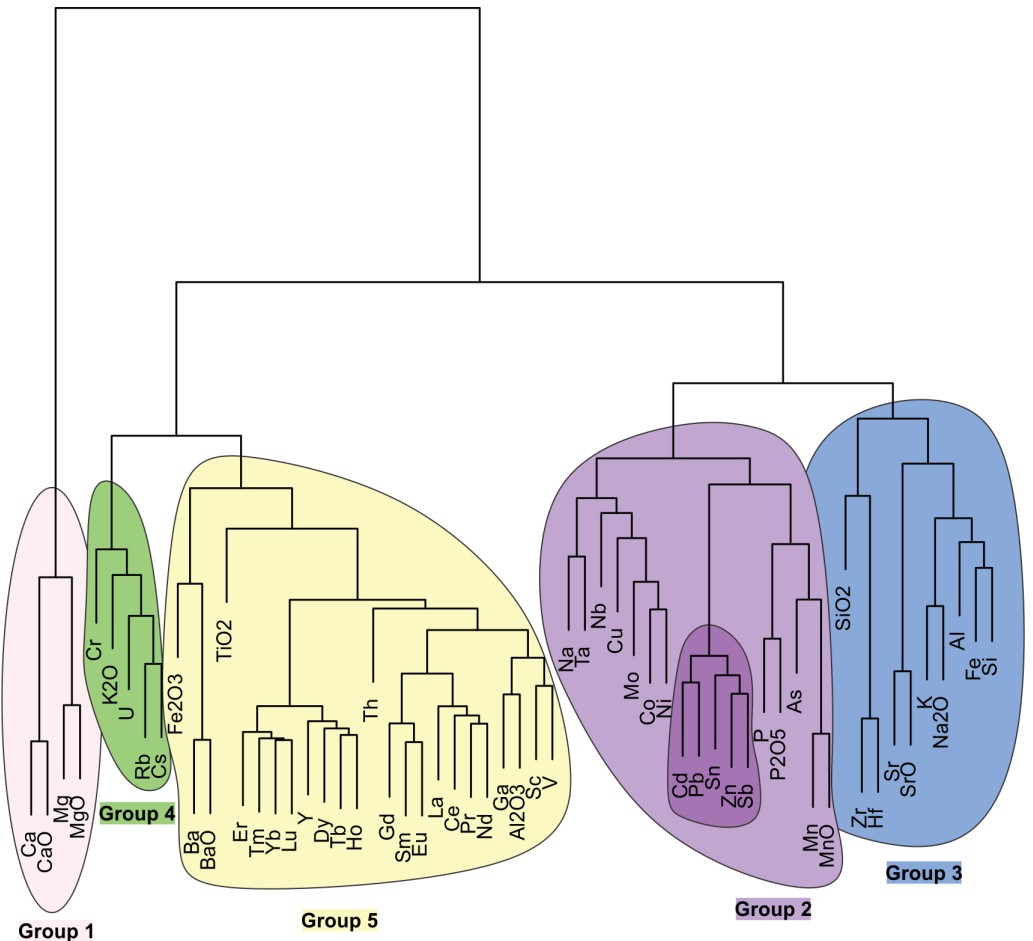

**Figure 3** Dendrogram of cluster analysis for 60 analytes in the Great Lakes sediment cores. Li and Ti are excluded from this analysis due to missing values. Five groups were defined based on the dendrogram and were numbered based on their positions in the PCA (Fig. 4).

*Christensen & Osuna, 1989*; *Biegalski & Hopke, 2004*; *Yohn et al., 2004*; *Shotyk, Krachler & Chen, 2005*; *Norton et al., 2007*; *Ettler et al., 2010*).

- Group 3 analytes are mostly common crustal elements with sources such as soil dust, runoff, and coastal erosion. Sr, K, and Fe are also related to mining inputs. Iron is mobile in sediments and can concentrate just below the sediment/water interface due to redox-related migration, a phenomenon that is visible as a rust colored sediment layer in Lake Superior cores (*Shaw Chraïbi et al., 2014*).
- Group 4 contains analytes which are widely distributed but are magnified by anthropogenic inputs. U and Cs are both associated with nuclear power and weapons testing (*Alloway, 2013*; *Reavie et al., 2005*). Ru and U are associated with coal combustion (*Campbell et al., 2005*; *Alloway, 2013*). Cr is associated with steel production and leather tanning (*Reavie et al., 2005*).
- Group 5 is the largest and most diverse group containing all of the rare earth elements. Some other elements included are Ba, which is used as a drilling fluid (*Moore, 1991*),

and Th, which is a byproduct of uranium mining (*ATSDR, 1990*). Otherwise, most of these elements are not known to be directly anthropogenically sourced though they may be increased due to erosion.

## PCA

Based on PCA (Fig. 4), 51% of the variance in the geochemical data is explained by the first two axes, which we present visually. The first, second, and third axes described 38.5%, 12.8%, and 8.2% of the variance, respectively. The five groups of analytes identified by cluster analysis are clearly segmented along the first axis. We infer this primary gradient to largely represent the bedrock and soil conditions in the vicinity of the sediment cores, indicating that natural setting is the strongest determinant of geochemical condition across the Great Lakes basin. The second axis appears to be a measure of contamination and temporal changes within cores, with more anthropogenically enriched analytes (e.g., Zn, Cd, Sn, Sb, Pb; Group 2a) having higher axis 2 values. Figure 4B explores the temporal trajectories of sample scores. While not clearly apparent for all cores, there is a general pattern of positive movement along axis 2 from the oldest to most recent samples, indicating movements toward the contaminant Group 2a. Additional PCA score trajectory data are used in conjunction with each lake's results (below).

## Geochemical profiles

Figure 5 through 11 present the geochemical profiles of each grouped cluster of analytes. These results are mainly used for lake-specific interpretations, but some basin-wide observations in historical trends are notable. Carbonate-related analytes were variable or monotonous since ~1850 (Fig. 5), but Lake Ontario had a distinct peak in Ca and CaO centered around the 1970s. Concentrations of Group 2a (Fig. 6; heavy metal contaminants) increased starting in the early 1900s, peaking around 1970. The Lake Ontario cores have the largest increases in these Group 2a elements followed by Lake Michigan. Concentrations of the Group 2 analytes (Fig. 7) followed erratic profiles, however there are a few exceptions. Manganese, MnO, and As concentrations have near-surface peaks likely related to redox conditions in the sediment. All cores increase in Na concentrations from presettlement times, accelerating quickly beginning around a 20-year period from the 1950s–70s with surface peaks in many cores. The rise in Na began much earlier in the 20th century in western Lake Superior.

Group 3 analytes (Fig. 8), representing terrigenous sediment inputs, also show few consistent patterns over time. However, Sr and SrO peaked in Lake Ontario between 1979 and 1984. Al concentrations decreased over time in Lake Huron North. Several cores had near-surface peaks in Fe likely related to redox activity in the sediments. Throughout the period of analyses the composition of instrumental calibration standards changed, preventing accurate measurement of Li or Ti in certain cores, so they were not part of multivariate analyses; they are plotted with this group due to similar profiles. Lithium concentrations increased over time for Lake Huron North and decreased for Superior West, Michigan South, and Huron South. Lake Superior West and the Lake Superior mid-lake "Isle Royale" core had very similar Ti profiles, with jagged peaks in the 1980s and 1990s and then decreases to surface intervals.

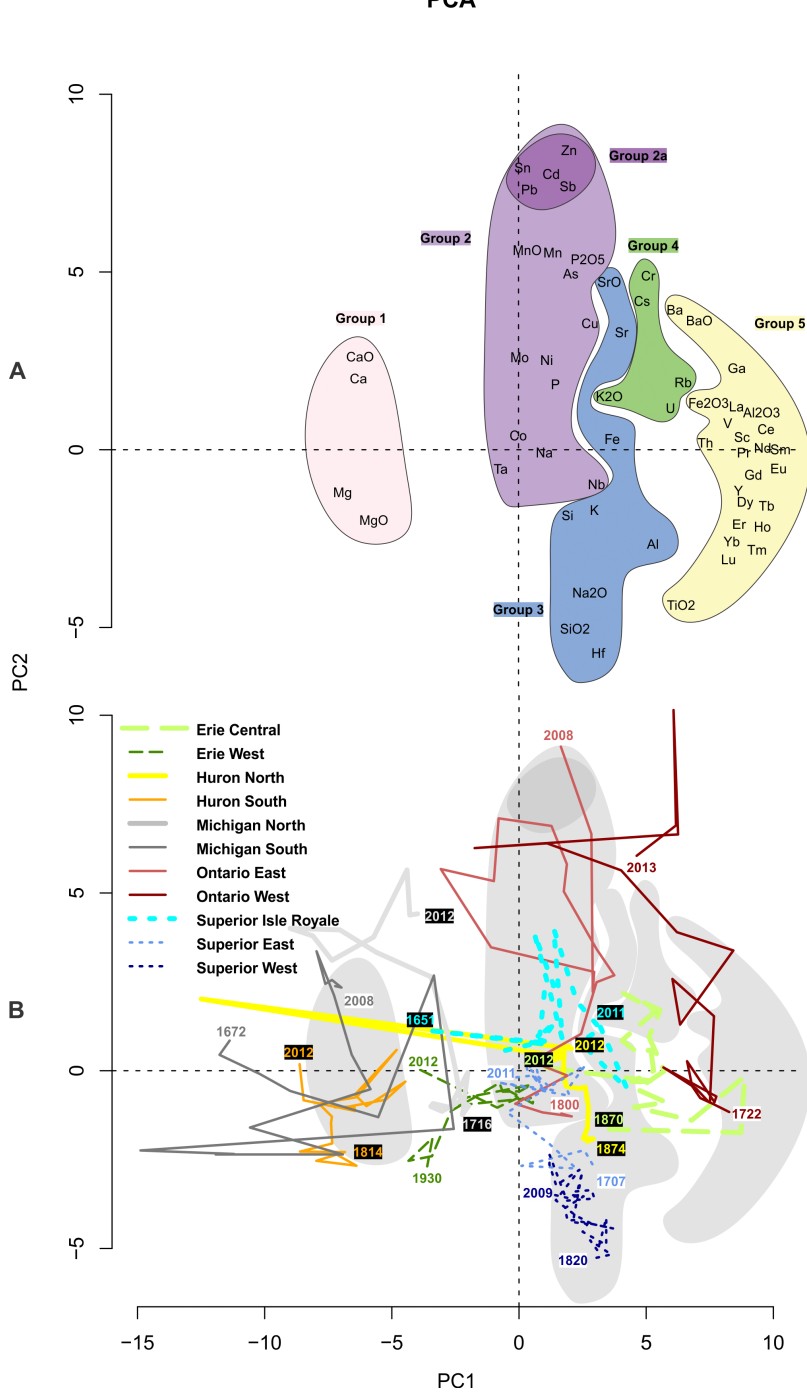

**Figure 4 First two axes of a principal components analysis of all sediment sample metals geochemistry.** (A) Analyte scores with the five groups (as identified by cluster analysis; Fig. 3) outlined and filled in with different color. (B) Sample score trajectories for each core. Grey shapes in the background represent the groups shown in (A).

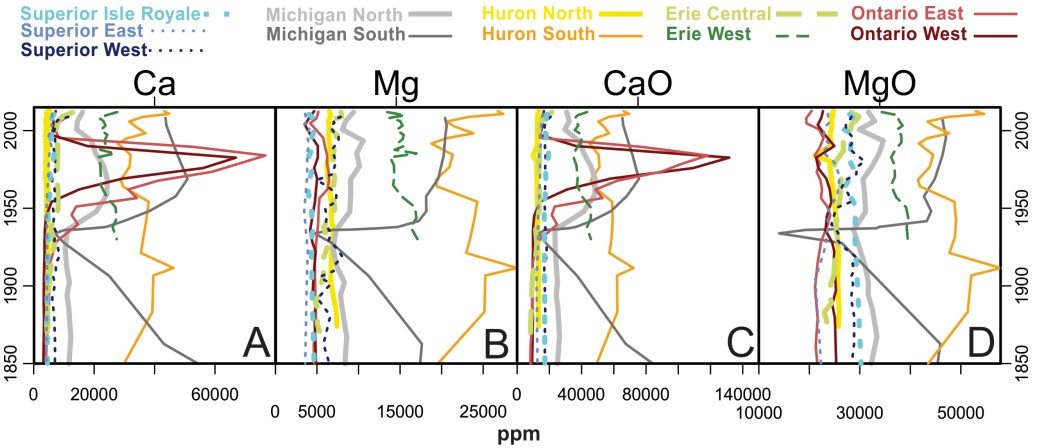

**Figure 5** **Analytes defined as Group 1 by cluster analysis.** These analytes are common carbonates from terrigenous clay minerals and carbon cycling in lakes.

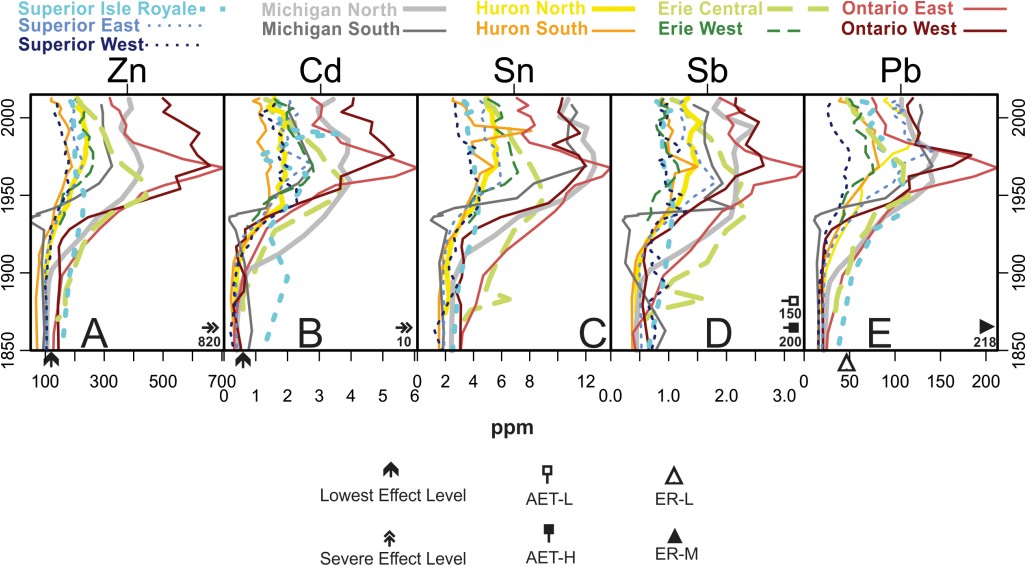

**Figure 6** **Analytes defined as group 2a by cluster analysis.** These analytes are recognized as distinct heavy metal contaminants in the Great Lakes system and are all known atmospheric contaminants. Arrows represent lowest and severe effect level values according to Ontario Sediment Quality Guidelines. For Pb no values were available using Ontario guidelines so ER-L (open triangle) and ER-M (closed triangle) values (US toxicity guidelines) are shown. For Sb neither of the two other indices had guidelines so AET-L (open square) and AER-M (black square) values (USEPA guidelines from Puget Sound, Washington) were used.

The concentrations of Group 4 analytes associated with coal ash and nuclear power generation (Fig. 9) are mostly stable over time, though $K_2O$ densities decreased in Lakes Ontario and Michigan. Cr had a spike in Lake Michigan North in 1867 and Lake Ontario West in 1928.

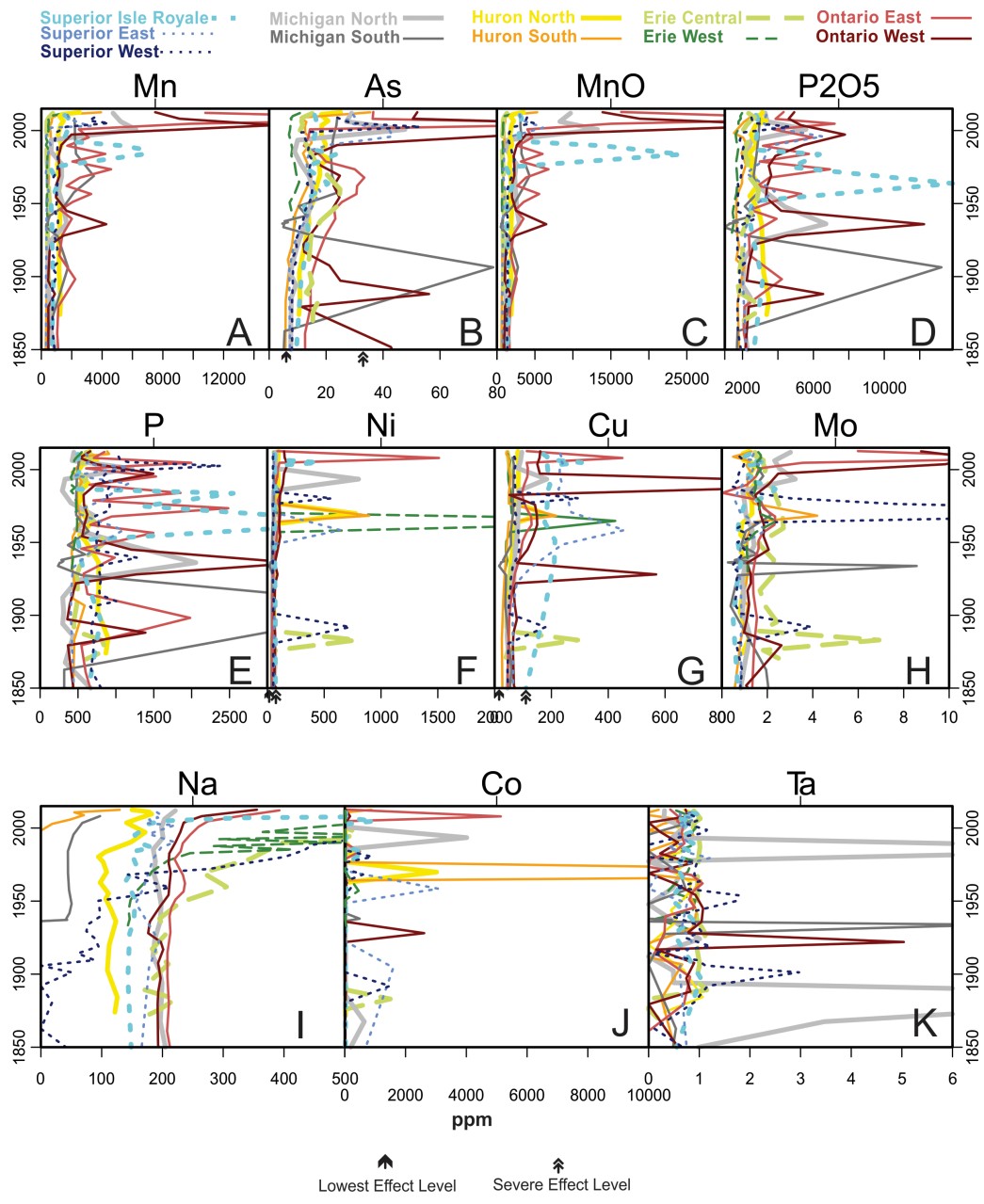

**Figure 7 Analytes defined as Group 2 by cluster analysis.** Analytes from diverse natural sources, though some are considered anthropogenic pollutants. Arrows represent lowest and severe effect level values according to Ontario Sediment Quality Guidelines.

Figure 10 represents Group 5's rare earth and associated element concentrations over time. Many of the profiles are monotonous and erratic, but there appears to be a slight downward trend over time, and there is a distinctive dip in each of these elements in the Michigan South core in 1933. Lake Ontario cores and the Lake Michigan south core had recent peaks in Ba and BaO.

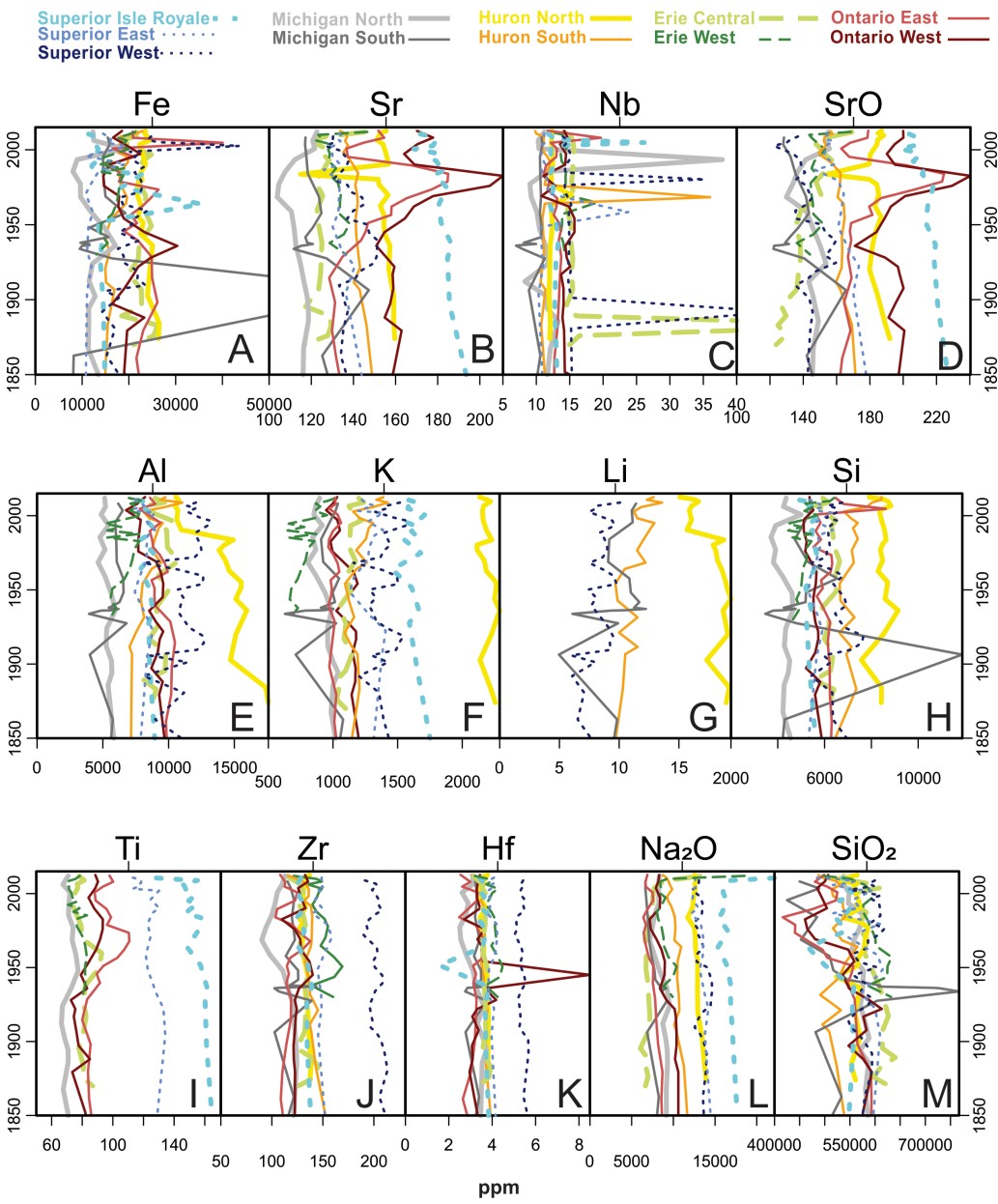

**Figure 8  Analytes defined as Group 3 by cluster analysis.** Common crustal analytes with lake sources such as soil dust, runoff, and coastal erosion. Li and Ti had too many NA values to be part of the PCA or cluster analyses but were grouped with Group 3 elements based on similar profiles.

## Lake Superior

Lake Superior samples are mainly associated with the Group 3 (siliciclastic, watershed-derived) analytes (Fe, Sr, K, Na₂O, Si), though the Isle Royale core is specifically influenced by Group 2 analytes such as Mn, P, and their oxides (Figs. 4, 7 and 8). The core sample scores track positively along PC2, with Superior West as the least enriched with Group 2 metals and Isle Royale as the most enriched. Upward migrations along PC2 (Fig. 4) reflect

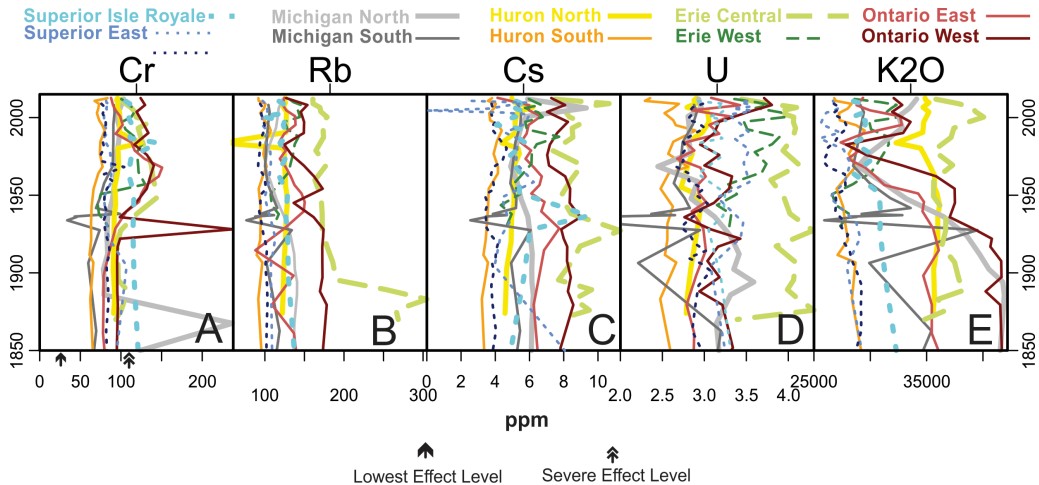

**Figure 9  Analytes defined as Group 4 by cluster analysis.** These analytes are considered widely distributed but magnified by anthropogenic inputs. Arrows represent lowest and severe effect level values according to Ontario Sediment Quality Guidelines.

increases in Group 2a contaminants such as Pb and Cd during the mid-20th century, with declines in these metals after the 1950s (Fig. 6). Of all the cores in this study, Lake Superior West had the tightest sample score distribution, indicating the smallest quantitative change in geochemical characteristics over time.

A gradual rise in Na concentration (Fig. 7) is evident in the 20th century through the uppermost sediments in all Superior cores, though the western core, which is closest to the urban centers of Duluth and Superior, has the earliest and most substantial rise, especially after the 1950s. Lake Superior cores had a distinct peak in metals just below the sediment surface denoted by peaks in Mn, As (Fig. 7) and Fe (Fig. 8) corresponding with a known horizon formed by the migration of redox-induced mobilization of metals (*Li et al., 2012*). Several Group 3 (Fig. 8) and Group 5 (Fig. 10) analytes gradually declined in concentration over the last ~100 years.

Cu in Superior East began to increase in concentration around 1922 from less than 100 ppm, peaking around 1958 at 456 ppm and decreasing to ~220 ppm by the 1970s, after which it has remained stable. The installation of new mines in Lake Superior's catchment peaked between 1900 and 1920 (Fig. 2; *Kerfoot et al., 1999*), well before this notable rise in Cu. Superior West had peaks in Ni and Cu in 1980.

## Lake Michigan

The Lake Michigan cores were both associated with the Group 1 (carbonate) analytes (Ca, Mg, CaO, MgO) though the Lake Michigan South core had a dip in all of these analyte concentrations beginning around 1948 and ending around 1993 when the analytes increased to well over historic levels (Figs. 4 and 5). Both cores tracked positively along PC2 over time. The northern core had stronger increases along PC2 over time and the Lake Michigan South core had less net change (Fig. 4). Upward migrations along PC2 reflect increases in Group 2a contaminants such as Sn and Pb during the mid-20th century, with

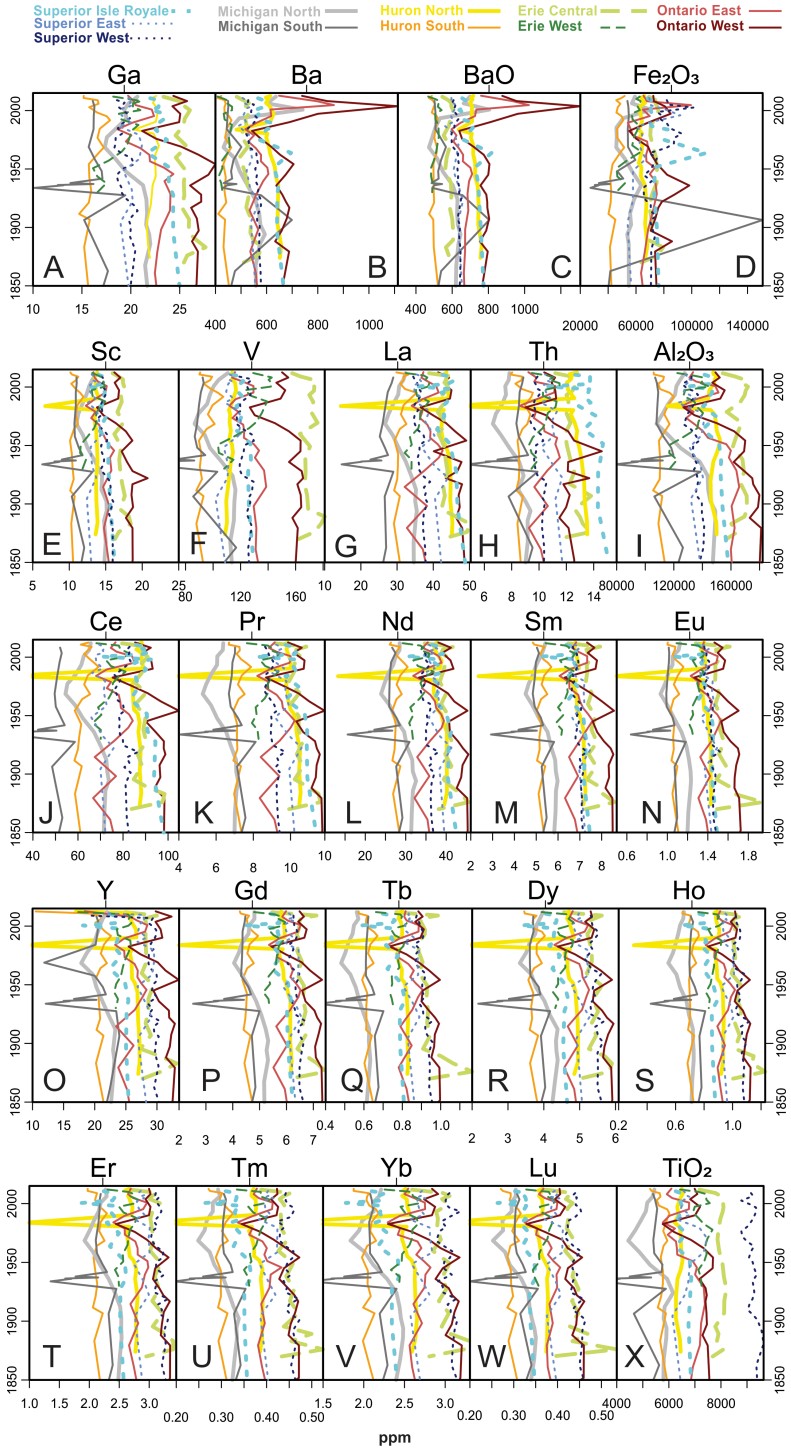

**Figure 10 Analytes defined as Group 5 by cluster analysis.** This is the largest and most diverse group which contains all the rare earth elements.

declines in these metals after the 1960s (Fig. 6). Group 2a analyte concentrations in the south core of Lake Michigan were second only to Lake Ontario, indicating substantial enrichment in contaminants in closer proximity to urban centers such as Chicago. The rise

in Na was more subtle in Michigan compared to other lakes (Fig. 7), though the southern core demonstrated a distinct horizon of first Na detection around 1940. The Lake Michigan South core experienced dips in Group 5 analytes (especially the rare earth elements) around 1933 potentially representing a large sedimentation event that would have diluted sediment metals (Fig. 10). Both cores experienced a less extreme dip in these analytes around 1968. In recent decades both cores have had drops in Group 2a contaminants, though Pb and Sn are now higher in Michigan South than any other Great Lakes core (Fig. 6). There have been drops in the Group 1 carbonate elements to around pre-impact levels (Fig. 5). In Michigan North, there have been recent increases in Group 5 elements (Fig. 10) and the Group 4 elements Cr, Rb, and $K_2O$ (Fig. 10).

## Lake Huron

Lake Huron South was primarily associated with Group 1 (carbonate) analytes (Fig. 4), having the highest stable concentrations of all of these elements and the highest concentrations of Mg and MgO, both historically and in modern times (Fig. 5). Huron North was associated primarily with the terrestrial analytes of Group 3 and had the highest stable concentrations of Al, K, and Si, though Al concentrations decreased steadily over time (Figs. 4 and 8). Both cores tracked positively along PC2 over time, though they both had relatively tight distributions of sample scores (Fig. 4). These increases reflect increasing concentrations of Group 2a concentrations such as Sn and Pb, which both peaked in the 1990s (Fig. 6). Lake Huron cores had peaks in many Group 2 elements (Fig. 7), corresponding with a horizon likely formed by the migration of redox-induced mobilization of metals (e.g., Mn, As, Fe; Fig. 8). Lake Huron South historically had Na concentrations below detection limits until the most recent decade, when Na increased to 130 ppm by 2012 (Fig. 7). Prior to the 1970s, the northern location contained Na concentrations around 110 ppm, after which Na levels increased. Lake Huron South had a peak in Nb around 1968 (Fig. 8). The Lake Huron South core has gradually decreased in Group 5 (rare earth) analyte concentrations in recent times (Fig. 10). Around 1983, the Lake Huron North core had a notable dip in all of the Group 5 analytes (Fig. 10) along with Sr and SrO (Fig. 8).

## Lake Erie

Lake Erie's central basin was primarily associated with the terrestrial analytes of Group 3, especially Fe and $SiO_2$ (Figs. 4 and 8). Lake Erie's western basin has been associated with analytes between the carbonates of Group 1 and the terrestrial elements of Group 3 such as Mg, Zr, and $SiO_2$. Both cores moved up PC2 slightly over time (Fig. 4), representing increases in Group 2a (contaminant) analytes (Fig. 6). Unlike other lakes, historical trends in carbonate-related analytes show a gradual decline but are otherwise unremarkable (Fig. 5). These increases were smaller than those of Lakes Michigan and Ontario but larger than those of Superior and Huron. The central core had a peak in many elements including Sn, Sb, Cu, Mo, Na, Co, and Nb in the 1880s (Figs. 6, 7 and 8). A gradual rise in Na concentration (Fig. 7) is evident in the 20th century through the uppermost sediments in both Erie cores, though the central core, which is adjacent to the urban center of Cleveland,

began its rise before the 1950s. There was a hump in As concentrations in the Erie West core in the late 1960s (Fig. 7). For Group 2 analytes, Ni and Cu both had peaks in the 1960s (Fig. 7). Figure 9 reveals a rise and fall of Cr during the mid-20th century as well as a fairly persistent rise in U and $K_2O$ in western Erie. As for other lakes, a gradual decline in rare earth elements is apparent for Lake Erie over the last ~100 years (Fig. 10).

## Lake Ontario

Lake Ontario East was historically associated with the soil derived analytes of Group 3 (Figs. 4 and 8), specifically Si. Lake Ontario West was historically related to the rare earth analytes of Group 5 (e.g., Ce, Pr, Nd, Sm, Eu, Gd, Tb, and Dy; Figs. 4 and 10). Lake Ontario had the highest increases in PC2, the dimension associated with pollutants (Fig. 4) related to the cores having the highest peaks in all Group 2a analytes (Zn, Cd, Sn, Sb, and Pb; Fig. 6). Lake Ontario West had a peak in Cr around 1928 (Fig. 9). The Lake Ontario west core also had a peak in P concentrations in the late 1940s (Fig. 7). Notable Ca and CaO peaks occurred from ~1960 through ~1990 (Fig. 5), and there was a hump in As concentrations in the late 1960s (Fig. 7). The highest Na concentrations occurred near the surfaces of the cores (Fig. 7). Lake Ontario also had peaks around 1980 in several other elements including Sr and Ba (Figs. 8 and 10), while most of the other rare earth elements exhibited a decline in the last few decades (Fig. 10). In the late 1960s and early 1970s, the Ontario East core had elevated levels of Ti (Fig. 8). Cores had a distinct peak in metals just below the sediment surface denoted by peaks in Mn, As (Fig. 7) and Fe (Fig. 8).

## Historical contaminant enrichment

Pre-post-impact transition dates were defined as follows using historical stressor data (Fig. 2): Lake Superior = 1900 due to a clear horizon of new mining; Lake Michigan = 1880 due to an increase in agriculture and mining; Huron = 1880 due to an agricultural increase; Erie = 1880 due to increases in agriculture and to correspond with dates confirmed by *Yuan (2017)*; and Ontario = 1900 due to increases in agricultural land and decreases in forest land.

For the purposes of calculating these two indices, Group 2 and 2a elements (determined by cluster analysis and PCA) were considered the contaminant elements of interest. $I_{geo}$ and EF values were placed on a heatmap (Fig. 11), highlighting the contaminant element that had the maximum value for each of the indices in a given decade. The contamination indices $I_{geo}$ and EF showed similar temporal trends to one another though there were some differences. $I_{geo}$ indices never achieved more than a "heavily" contaminated ranking (most were considered uncontaminated or moderately contaminated), whereas EF had several cases indicating "extreme" enrichment of contaminant elements, especially for Co.

In Lake Superior, Na was the most common, dominant contaminant since the beginning of the 20th century, well before road salt was in common use. The most contaminated decade in Lake Superior was the 1950s with an $I_{geo}$ of 0.2 and a 3.6 EF value with Co being the most enriched element. The 1980s ($I_{geo}$ of 0.2, EF of 2.1), 2000s ($I_{geo}$ of 0.2, EF of 2.6), and 2010s ($I_{geo}$ of 0.3, EF of 1.7) were also contaminated, with Na being the maximum value except for EF in 2010, which was dominated by Mn.

| | Superior | | | | Michigan | | | | Huron | | | | Erie | | | | Ontario | | | |
|---|---|---|---|---|---|---|---|---|---|---|---|---|---|---|---|---|---|---|---|---|
| | $I_{geo}$ Avg | $I_{geo}$ Max | EF Avg | EF Max | $I_{geo}$ Avg | $I_{geo}$ Max | EF Avg | EF Max | $I_{geo}$ Avg | $I_{geo}$ Max | EF Avg | EF Max | $I_{geo}$ Avg | $I_{geo}$ Max | EF Avg | EF Max | $I_{geo}$ Avg | $I_{geo}$ Max | EF Avg | EF Max |
| 2010 | 0.3 | Na | 1.7 | Mn | 0.5 | Pb | 3.5 | Cd | 0.5 | Co | 3.0 | Co | 0.1 | Na | 2.7 | Na | 0.8 | Mn | 4.0 | Mn |
| 2000 | 0.2 | Na | 2.6 | Na | 0.7 | Pb | 3.2 | Mn | 0.2 | Pb | 1.9 | Pb | -0.2 | Ta | 1.3 | Na | 1.1 | Co | 8.8 | Co |
| 1990 | -0.1 | Na | 1.8 | Na | 0.9 | Co | 4.1 | Co | 0.3 | Pb | 2.2 | Pb | 0.0 | Co | 1.6 | Co | 0.5 | Cu | 3.0 | Cu |
| 1980 | 0.2 | Na | 2.1 | Na | 0.5 | Cd | 3.1 | Cd | 0.1 | Co | 1.7 | Co | -0.2 | Ta | 0.5 | Ta | 0.1 | Co | 3.1 | Co |
| 1970 | -0.1 | Na | 1.7 | Na | 0.2 | Cd | 2.8 | Cd | 0.3 | Pb | 1.8 | Pb | -0.1 | Cd | 1.2 | Cd | 0.4 | Cd | 3.2 | Cd |
| 1960 | 0.1 | Na | 1.8 | Mo | 0.6 | Mn | 3.2 | Cd | 0.8 | Co | 35.2 | Co | 0.0 | Cd | 1.3 | Cd | 0.5 | Cd | 2.8 | Cd |
| 1950 | 0.2 | Co | 3.6 | Co | 0.5 | Cd | 2.7 | Cd | 0.1 | Cd | 1.4 | Cd | 0.4 | Ta | 1.8 | Cd | 0.4 | Cd | 2.4 | Cd |
| 1940 | -0.1 | Cd | 1.9 | Cd | 0.3 | Cd | 2.0 | Cd | 0.1 | Cd | 1.4 | Cd | 0.3 | Ta | 1.6 | Cd | 0.2 | Cd | 2.1 | Cd |
| 1930 | -0.4 | Na | 1.0 | Na | -0.3 | Co | 1.9 | Co | 0.0 | Cd | 1.2 | Cd | 0.1 | Ta | 1.4 | Cd | 0.2 | Mn | 0.1 | Mn |
| 1920 | -0.5 | Na | 0.8 | Na | -0.3 | Cd | 1.4 | Cd | -0.3 | Cd | 1.0 | Pb | -0.1 | Ta | 1.1 | Ta | -0.2 | Co | -0.2 | Co |
| 1910 | -0.7 | Na | 1.1 | Na | -0.5 | Cd | 1.3 | Cd | -0.4 | Pb | 0.8 | Pb | -0.4 | Sb | 0.9 | Sb | -1.1 | Pb | 0.7 | Pb |
| 1900 | -0.5 | Co | 2.2 | Co | -0.3 | As | 2.0 | As | -0.1 | Co | 5.1 | Co | -0.5 | Sb | 0.7 | Sb | -0.9 | Ta | 0.5 | Sn |
| 1890 | | | | | -0.9 | Pb | 0.9 | Pb | 0.7 | Co | 15.4 | Co | -0.3 | Ta | 0.8 | Ta | | | | |
| 1880 | | | | | -0.6 | Ta | 1.6 | Ta | -0.4 | Ta | 0.8 | Co | 0.1 | Co | 3.8 | Co | | | | |

| | |
|---|---|
| $I_{geo}$ <0 | Uncontaminated |
| 0< $I_{geo}$ <1 | Uncontaminated to moderately contaminated |
| 1< $I_{geo}$ <2 | Moderately contaminated |
| 2< $I_{geo}$ <3 | Moderately to heavily contaminated |
| 3< $I_{geo}$ <4 | Heavily contaminated |
| 4< $I_{geo}$ <5 | Heavily to extremely contaminated |
| $I_{geo}$ >5 | Extremely contaminated |

| | |
|---|---|
| EF <2 | Deficiency to minimal enrichment |
| 2< EF <5 | Moderate enrichment |
| 5< EF <20 | Significant enrichment |
| 20< EF <40 | Very high enrichment |
| EF >40 | Extremely high enrichment |

**Figure 11** $I_{geo}$ **and EF heat map for each of the five lakes.** The numbers represent overall contamination values (Eqs. (1) and (2)) averaged for each of the "Group 2" contaminant elements of interest in the decade shown. The elements listed represent the element with the highest contamination values for the interval.

In Lake Michigan, sediments have been enriched with Group 2 elements consistently since the 1940s, being dominated by Cd in most decadal intervals. The highest $I_{geo}$ (0.9) and EF (4.1) values were in the 1990s with Co being dominant in both cores. After that, Pb, Cd and Mn were the dominant contaminants.

In Lake Huron, Co was a leading contaminant in the 1890s, 1900s, 1960s, and 2010s, with maximum $I_{geo}$ (0.8) and EF (35.2) values in the 1960s. Cd and Pb were the other leading contaminants in this lake beginning in 1910 all the way up until the 2000s.

Lake Erie $I_{geo}$ and EF values were strongly influenced by Ta and Cd, though the highest values in recent times were Na, with an average $I_{geo}$ of 0.1 (minimal contamination) and 2.7 EF (moderate enrichment).

Lake Ontario had the highest enrichment in the 2000s with an $I_{geo}$ of 1.1 (extremely contaminated) and a moderate EF of 8.8 attributed to high Co. Ontario also had high enrichment of Mn, Cd, and Cu.

## Contamination and toxicity guidelines

In Group 2a, there were guidelines for Zn, Cd, Sb, and Pb. For Zn, no cores were ever above the severe effect level, though lowest effect level exceedances were common. Lake Ontario, Lake Erie Central, and Lake Superior Isle Royale exceeded lowest effect level in pre-impact sediments. For Cd, Lake Michigan South and Lake Superior Isle Royale were

above the lowest effect level pre-impact, but no sediments exceeded the severe effect level. Sb did not have levels defined by the Ontario guidelines, so AET-L and AET-H levels were used. None of the cores were above either of these levels at any point. No cores had Pb levels above ER-M at any point, though every core except Lake Superior West exceeded ER-L at some point. Though levels have declined with time, all but Superior West and Huron South remain above ER-L for Pb.

In Group 2, there were guidelines for Mn, As, Ni, and Cu. For Mn, Lake Ontario East was at the severe effect level pre-impact. Lake Ontario West, Lake Michigan North, Lake Superior Isle Royale, and Lake Superior West were all at the lowest Mn effect level pre-impact. All lakes exceeded the severe effect level for Mn in recent sediments. For As, Lake Ontario West was above severe effect level pre-impact, and all cores except Huron North and Michigan South were above lowest effect level in pre-impact sediments. Both Ontario cores and Lake Huron South were above severe effect level for As in recent sediments. All cores in this study exceeded the lowest effect level for Ni in pre-impact sediments; Ontario West was above severe effect levels pre-impact. In recent sediments, both Ontario cores and Huron North were above Ni severe effect levels. For Cu, all core pre-impact sediments exceeded the lowest effect range; Lake Superior Isle Royale exceeded the severe effects level. In recent sediments, both Lake Ontario cores, Lake Superior Isle Royale, and Lake Superior East were above severe effects level for Cu.

Contaminant guidelines existed for Cr in Group 4. In pre-impact sediments, all cores were above the lowest effects level for Cr, while Michigan North and Superior Isle Royale were above severe effect levels. In recent sediments Erie, Ontario West, and Superior Isle Royale were above the severe effects level.

## Metal and stressor correlations

Several metals and oxides were strongly related to stressors. Group 1 (carbonate-related) analytes were strongly positively correlated with agricultural area and population while having negative correlations with mining stress and forestry (Table 1). Group 2a elements, the well-known atmospheric contaminants (*Norton et al., 2007*; *Ettler et al., 2010*), had strong positive relationships with population and weaker, negative correlations with mine stress. Pb had a positive correlation with forestry. The remaining Group 2 elements had few correlations with stressors, though some (Na, Mo, Ni, As, Mn, and MnO) had some affinity with population. Nb was negatively correlated with forestry, and Cu was negatively correlated with agricultural area. Group 3 (common crustal) analytes were largely negatively correlated with agricultural area and population and several were positively correlated with forestry and mine stress. Some Group 4 analytes (a complicated set of potentially anthropogenic pollutants) were positively correlated with agricultural area ($K_2O$, Rb, Cs) and population (Cr, U, Rb, Cs), while Rb and Cs had strong negative correlations with forestry and mine stress. Group 5, the widespread rare earth analytes, had mainly negative correlations with agricultural area and, to a lesser extent, forestry and population. $TiO_2$, Er, Tm, Tb, Lu, Y, and Dy had weak but significant correlations with mine stress.

**Table 1 Pearson correlations of stressors against downcore analyte data.** Analytes are grouped as determined by cluster analysis (Fig. 3). Coloring applies to positive (red) and negative (blue) relationships and asterisks reflect three levels of $P$-value.

| Group | Analyte | Ag. Area | | Forestry | | Mine stress | | Population | |
|---|---|---|---|---|---|---|---|---|---|
| 1 | Ca | 0.37 | *** | −0.25 | ** | −0.28 | *** | 0.4 | *** |
| | CaO | 0.32 | *** | −0.19 | ** | −0.23 | ** | 0.36 | *** |
| | Mg | 0.42 | *** | −0.31 | *** | −0.24 | ** | 0.28 | *** |
| | MgO | 0.32 | *** | −0.23 | ** | −0.17 | ** | 0.16 | * |
| 2a | Cd | 0.01 | | 0.09 | | −0.16 | * | 0.52 | *** |
| | Pb | −0.03 | | 0.29 | *** | −0.16 | * | 0.37 | *** |
| | Sn | 0.25 | ** | −0.04 | | −0.22 | ** | 0.63 | *** |
| | Zn | 0.1 | | −0.05 | | −0.24 | ** | 0.55 | *** |
| | Sb | 0.1 | | 0.05 | | −0.13 | * | 0.46 | *** |
| 2 | Na | −0.01 | | −0.09 | | −0.12 | | 0.2 | ** |
| | Ta | −0.03 | | 0.07 | | 0.03 | | −0.03 | |
| | Nb | −0.1 | | −0.17 | ** | 0.04 | | −0.03 | |
| | Cu | −0.29 | *** | 0.16 | * | −0.04 | | −0.04 | |
| | Mo | −0.06 | | −0.11 | | −0.08 | | 0.17 | * |
| | Co | 0 | | 0 | | −0.01 | | −0.01 | |
| | Ni | 0.05 | | −0.04 | | −0.04 | | 0.16 | * |
| | P | −0.13 | | 0.03 | | 0.01 | | −0.09 | |
| | $P_2O_5$ | −0.08 | | 0.14 | * | −0.1 | | 0.06 | |
| | As | −0.14 | * | 0.01 | | −0.17 | * | 0.14 | * |
| | Mn | −0.11 | | 0.1 | | −0.1 | | 0.23 | ** |
| | MnO | −0.13 | | 0.1 | | −0.1 | | 0.21 | ** |
| 3 | $SiO_2$ | 0.14 | * | −0.1 | | 0.17 | * | −0.17 | * |
| | Zr | −0.51 | *** | −0.11 | | 0.42 | *** | −0.36 | *** |
| | Hf | −0.43 | *** | −0.11 | | 0.39 | *** | −0.33 | *** |
| | Sr | −0.49 | *** | 0.15 | * | −0.15 | * | −0.22 | ** |
| | SrO | −0.34 | *** | 0.27 | *** | −0.18 | ** | −0.14 | * |
| | K | −0.58 | *** | 0.63 | *** | 0.27 | *** | −0.59 | *** |
| | $Na_2O$ | −0.68 | *** | 0.46 | *** | 0.33 | *** | −0.63 | *** |
| | Al | −0.5 | *** | 0.2 | ** | 0.23 | ** | −0.45 | *** |
| | Fe | −0.07 | | −0.14 | * | −0.09 | | −0.05 | |
| | Si | −0.3 | *** | 0.2 | ** | 0.08 | | −0.32 | *** |
| | Li | 0.15 | | 0.78 | *** | −0.12 | | 0.04 | |
| | Ti | −0.8 | *** | 0.54 | *** | 0.05 | | −0.58 | *** |
| 4 | Cr | −0.04 | | −0.04 | | −0.19 | ** | 0.3 | *** |
| | $K_2O$ | 0.4 | *** | −0.07 | | −0.11 | | 0.08 | |
| | U | 0 | | −0.21 | ** | −0.16 | * | 0.26 | ** |
| | Rb | 0.22 | ** | −0.32 | *** | −0.3 | *** | 0.18 | ** |
| | Cs | 0.25 | ** | −0.31 | *** | −0.35 | *** | 0.32 | *** |
| | $Fe_2O_3$ | −0.44 | *** | −0.02 | | 0.04 | | −0.2 | ** |
| | Ba | −0.3 | *** | 0.19 | ** | 0.09 | | −0.12 | |

| Group | Analyte | Ag. Area | | Forestry | | Mine stress | | Population | |
|---|---|---|---|---|---|---|---|---|---|
| | BaO | −0.32 | *** | 0.19 | ** | 0.06 | | −0.12 | |
| | TiO$_2$ | −0.45 | *** | −0.3 | *** | 0.25 | ** | −0.22 | ** |
| | Er | −0.38 | *** | −0.24 | ** | 0.18 | ** | −0.18 | ** |
| | Tm | −0.38 | *** | −0.3 | *** | 0.13 | * | −0.17 | * |
| | Yb | −0.33 | *** | −0.28 | *** | 0.19 | ** | −0.15 | * |
| | Lu | −0.35 | *** | −0.29 | *** | 0.17 | ** | −0.16 | * |
| | Y | −0.39 | *** | −0.2 | ** | 0.15 | * | −0.2 | ** |
| | Dy | −0.4 | *** | −0.22 | ** | 0.16 | * | −0.18 | ** |
| 5 | Tb | −0.45 | *** | −0.26 | *** | 0.06 | | −0.19 | ** |
| | Ho | −0.45 | *** | −0.27 | *** | 0.11 | | −0.2 | ** |
| | Th | −0.42 | *** | 0.1 | | −0.12 | | −0.25 | ** |
| | Gd | −0.43 | *** | −0.16 | * | 0.09 | | −0.2 | ** |
| | Sm | −0.41 | *** | −0.14 | * | 0.01 | | −0.19 | ** |
| | Eu | −0.39 | *** | −0.17 | ** | 0.04 | | −0.19 | ** |
| | La | −0.38 | *** | 0.04 | | −0.05 | | −0.18 | ** |
| | Ce | −0.4 | *** | −0.04 | | −0.01 | | −0.16 | * |
| | Pr | −0.48 | *** | −0.08 | | −0.09 | | −0.23 | ** |
| | Nd | −0.42 | ** | −0.05 | | −0.01 | | −0.21 | ** |
| | Ga | −0.17 | * | −0.03 | | −0.1 | | −0.07 | |
| | Al$_2$O$_3$ | −0.14 | * | −0.1 | | −0.06 | | −0.12 | |
| | Sc | −0.22 | ** | −0.24 | ** | 0.01 | | −0.08 | |
| | V | −0.13 | | −0.4 | *** | −0.18 | ** | 0.09 | |

Notes.
*$P < 0.05$
**$P < 0.01$
***Bonferroni correction for $P = 0.05$

# DISCUSSION

Natural, lake-specific physicochemical conditions and unique anthropogenic histories likewise necessitate lake-specific interpretations for geochemistry and contaminant history in the Great Lakes. However, certain patterns in ubiquitous metals and key pollutants were observed throughout the basin, reflecting widespread anthropogenic effects. One example is the up-core reduction in all rare earth elements (Group 5; Fig. 10), which may represent a long-term dilution of natural elements as human development intensified and the flux of other materials (organic and inorganic) increased. The most prevalent basin-wide pollutants were Pb, Cd, Co, and Na. Historical lead pollution is largely related to combustion of leaded gasoline, and since enactment of removal regulations in the 1970s the concentration of Pb has declined in all of the sediment cores (*Yuan, 2017*). Such long-term trends in this widely scattered, toxic pollutant have been observed worldwide in sedimentary records (e.g., *Farmer et al., 1996*; *Heyvaert et al., 2000*). According to *Sherman et al. (2015)*, after leaded gasoline was banned sources of Pb are likely atmospheric deposition from coal and oil combustion, metal mining and processing, battery recycling, and oil refining. Despite continued atmospheric releases, the gasoline ban clearly reduced atmospheric deposition of Pb to aquatic systems in North America. The other related (Group 2a)

elements—Cd, Sn, Zn, and Sb—follow patterns in historical Pb, as these atmospheric pollutants volatilize at high temperatures and are frequently associated with smelting and fossil fuels (*Norton et al., 2007*; *Ettler et al., 2010*). Cobalt is also an atmospheric contaminant, and one often associated with forest fires (*ATSDR, 2004*), so it is not surprising that it is found at enriched concentrations throughout the Great Lakes basin sediments. Cobalt has not been mined within the Great Lakes basin, but it is likely a byproduct of other mining activities due to its presence in many rock types (*Carr & Turekian, 1961*).

In the metal-stressor correlations (Table 1 and Fig. 2), there is clearly decoupling between agricultural area and population, and especially mines and populations. Landscape mine stress poorly represents itself in these deep lake cores, and instead we get erosional signals (such as seen in Group 3 analytes) during early settlement periods. In terms of sediment geochemistry, human population and sometimes agriculture are good tracers while mine stress is not, probably because actual mine waste is apparently not providing a useful signal in these deep lake cores, the one exception being records from Cu from Lake Superior, described below.

Salt pollution is also revealed as a prevailing phenomenon in the Great Lakes. In the United States, annual rock salt used for road deicing was 163,000 tons in 1940, increasing to more than 23 million tons in 2005 (*Novotny, Murphy & Stefan, 2008*). In Minnesota, rock salt use increased from 60,000 to 900,000 tons in the same time period. This rock salt has led to increased salinity in lakes near major roadways of urban watersheds. Sodium is unsurprisingly found at higher concentrations near the surfaces of cores as a result of its prevailing use as a deicing agent (NaCl) on roads since the late 1940s. The use of rock salt can be attributed to Na and Cl tracing each other in sedimentary and water quality datasets, and while increasing lakewater Cl concentrations have been linked concurrently with industrial inputs (*Sonzogni et al., 1983*), concurrence with sedimentary Na trends clarifies the importance of road salt. Lake Superior and Lake Michigan Cl concentrations have continued rising and are currently at maximum recorded levels. In Lakes Huron, Erie, and Ontario Cl concentrations peaked between 1965 and 1975 and then decreased, though recent data indicate Cl levels are increasing again (*Chapra, Dove & Rockwell, 2009*). Na levels in Lake Superior began to increase decades before road salt usage began in the 1940s, possibly because Na is a component of the silicate rocks that comprise the majority of its disturbed watershed (*Chapra, Dove & Warren, 2012*). In addition to road applications rising, Na values may also represent increases in runoff, weathering, and dissolution of these rocks and soils.

While most of the analytes considered in this manuscript tend to be stable in sedimentary records, some are subject to post depositional mobility, which may obscure our ability to use them for historical interpretations. For example, subsurface peaks in Mn, As (Fig. 6) and Fe (Fig. 7) observed across the Great Lakes likely correspond with a typical horizon formed by the migration of redox-induced mobilization of these metals (*Li et al., 2012*). While geochemically interesting, uncertainty should be applied when making historical interpretations from profiles of these mobile metals. Many metals are mobilized by these oxides, which may further complicate interpretation (*Presley, Trefry & Shokes, 1980*).

Expressing a Great Lakes basin-wide interpretation of historical geochemistry is challenging because of strong physical, chemical, and historical anthropogenic differences among the lakes, so it is worthwhile summarizing geochemical histories that are specific to each lake.

### Lake Superior

Lake Superior's confined distribution of sample scores according to the PCA (Fig. 4) probably relates to its low sediment accumulation rate and the lowest lake water ion concentrations (*Chapra, Dove & Warren, 2012*). Due to its relatively small human population (Fig. 2), Lake Superior has not received nutrient and industrial pollutant loads observed in other lakes (*O'Beirne et al., 2015*). *Nriagu et al. (1995)* estimated that around 60–80% of anthropogenic inputs to this lake are atmospheric. However, the bedrock in Lake Superior watershed is metal rich, and between 1850 and 1929 the Keweenaw district was the 2nd largest producer of Cu in the world. Even prior to Euro-American settlement, Cu mining activities by Native American communities had detectable effects on lake geochemistry (*Pompeani et al., 2015*). For much of the Anthropocene (1850–1968), several hundred million metric tons of mine tailings were dumped into the lake (*Kerfoot, Lauster & Robbins, 1994*), an event that is clearly present in the geochemical profile of the eastern core, where Cu concentration peaked in 1958, lagging behind the largest period of mine openings, showing the long term impact of mining on this system. The highest population density around Lake Superior is in the St. Louis River drainage, which is the 2nd largest inflow to Lake Superior (*O'Beirne et al., 2015*), just upgradient from the western core location (Fig. 1). The proximity to the urban centers of Duluth and Superior explains the high Na levels, which are probably mostly related to road salt use (*Anderson, Estabrooks & McDonnell, 2000*). Iron mining in Lake Superior and disposal of mining waste directly to the lake (*Minnesota Historical Society, 2012*) lead to increases in water and sediment Fe concentrations that make for distinct iron-laden layers in sedimentary records. Due to redox drivers, Fe and related mobile metals are maintained near the tops of sediment profiles and therefore do not match up with historical records of waste disposal.

### Lake Michigan

In terms of 20th-century contamination (largely Group 2a analytes; Fig. 6), Lake Michigan was similarly impacted to Lake Ontario, which is considered to be the most impacted of the Great Lakes (*Allan et al., 2013*). Industrial contamination is attributed to the influences of the major cities of Chicago and Milwaukee and their industrialized surroundings. Lake Michigan's southern basin has limited river inflows and so inorganic contaminants to the lake tend to be atmospheric inputs from urban centers (*Winchester & Nifong, 1971*). *Rossmann, Pfeiffer & Filkins (2014)* found Pb concentrations in Lake Michigan sediments rose after 1850 due to coal and gas combustion, rose more with industrial activity around WWII, and lowered after the Clean Air Act of 1970 and the phase-out of leaded gasoline. These trends largely match our long-term trends for Pb in our sediment cores. The distinct dip in Group 5 analytes in Lake Michigan South around 1933 (Fig. 10) is difficult to explain but is probably a result of a substantial depositional event resulting in dilution of these

ubiquitous rare earth materials. It is countered by very high concentrations of $SiO_2$. The feature also aligns with the timing of the Dust Bowl and may represent a large deposition of sediment from a dust storm in the Great Plains (*Cook, Miller & Seager, 2008*).

In addition to Pb, the last century of sediments in Lake Michigan are defined by Cd and Co enrichment. Cd and Co are both sourced into the environment through mining and smelting emissions, the electroplating industry, and other industrial sources (*Jaagumagi, 1993*; *Jeong & McDowell, 2003*).

## Lake Huron

Lake Huron had similar long-term trends in Group 2a contaminants as the rest of the Great Lakes basin, in many ways revealing a similar atmospheric contamination signature as Lake Superior. The Lake Huron North core contains higher contaminant concentrations despite being adjacent to lower human populations than the southern core (Fig. 1). This may be due to the influence of the Sudbury mining district located just upgradient. Cadmium is one of the key pollutants in the lake, which is also sourced from the Sudbury area (*Bagatto & Alikhan, 1987*). Cobalt was also a major contaminant in this lake, and according to *ATSDR (2004)* 76% of atmospheric Co input into Lake Huron was from natural sources and 24% was from anthropogenic sources. Anthropogenic sources include fossil fuel combustion, vehicular exhaust, copper and nickel mining and smelting, and Co-containing phosphate fertilizers. An example of a likely natural source of Co pollution was the "Fletcher Road Fire" in 1968 in Crawford and Kalkaska Counties in northern Michigan that burned 4,216 acres of trees (*Michigan Department of State Police, 2012*). This event corresponds with the Co peak in the Lake Huron South core along with peaks in Ni, Cu, and Mo. Reasons for the dip in Group 5 analytes around 1983 (Fig. 10) are unknown, as is the decline in Al over time (Fig. 8); no erosional signals (e.g., $SiO_2$) suggest a dilution effect.

## Lake Erie

The Lake Erie basin has consistently had the highest population and an amount of agricultural land in the catchment that is comparable to the much larger Lake Michigan (Figs. 1 & 2). As for most of the Great Lakes basin, Lake Erie's geochemical record contains contaminant tracers of industrial growth during World War II and the Korean conflict (*Förstner, 1976*). Peaks of Zn, Cd, Sn, and Pb around 1949 and later declining by 1978 track higher atmospheric levels resulting from combustion of fossil fuels, particularly the trend for the fuel additive Pb. Enrichment of Cd in the central basin of Lake Erie in the late 1940s relates to the growth of the Cleveland electroplating industry (*Förstner, 1976*). Declines of these metal pollutants are related to the stringent regulations passed in the 1970s, the ban of leaded gasoline and improvement in municipal sewage (*Yuan, 2017*). Antimony peaked earlier, around 1938, which is earlier than *Walters, Wolery & Myser (1974)*, who found increases in Sb between 1948 and 1953 that they attributed to industrial and diffuse sources. Despite reductions in contaminant metals, persistent urbanization of Lake Erie's catchment has maintained increasing concentrations of sediment Na. *Kemp et al. (1976)* found that approximately 60% of the heavy metal and nutrient loading to Lake Erie sediments was deposited in the Eastern Basin despite being sourced farther upstream

in the vicinities of Detroit and Cleveland, suggesting the contaminants were transported long distances. Closer geochemical investigation of Lake Erie's eastern basin is warranted.

## Lake Ontario

The Great Lakes Environmental Assessment and Mapping Project ranked Lake Ontario the most stressed of the five Great Lakes (*Allan et al., 2013*), a ranking that is geochemically supported by modern PCA scores being the most strongly associated with Group 2a contaminants (Fig. 3). Sedimentary contaminants were particularly high in our cores from Lake Ontario. Major sources of nutrients and toxic chemicals to Lake Ontario include cities along the Oswego, Genesee, and Niagara Rivers and metro drainages near Toronto, Kingston, and Ottawa (see the developed area in Fig. 1). There was also direct discharge of industrial and municipal waste from toxic waste dumps (e.g., Hyde Park and Love Canal) into the Niagara River (*Estepp & Reavie, 2015*). The Niagara River is the single largest tributary to Lake Ontario and is a major source for Cr, Cd, V, As, and a lesser but significant source of Pb, Cu, Ni, corresponding with especially high peaks of these metals in Lake Ontario (*Thomas, 1983*). In 1987, Canada and the United States signed a Niagara River Declaration of Intent aiming to reduce pollution, which corresponds to decreasing levels of these and other heavy metals in this system (Figs. 6, 7, 8 and 10; *Murphy, Bhavsar & Gandhi, 2012*). In addition to being heavily urbanized and industrialized, Lake Ontario is the farthest downstream in our study so it may also be integrating stressor information about the entire Great Lakes basin.

In Group 1 (Fig. 4), which contains carbonate minerals, Lake Ontario's historical association with carbonates is emphasized. Fluctuations in Ca compounds in this system are probably related to "whiting events", calcite precipitation during high primary productivity (algal blooms) during the summer months in the lake. *Hodell et al. (1998)* found an exponential rise in sediment carbonate after 1930, which peaked in the early 1980s, related to P loading that supported increased algal biomass. Though recycling of sedimentary P can make it an unreliable paleo-indicator (*Anderson, Rippey & Gibson, 1993*), the exponential rise in P concentrations in the late 1940s was likely due to increases in phosphate detergents in urban areas (*Estepp & Reavie, 2015*). Stratigraphic declines in Ca and CaO in Lake Ontario in the mid-1980s likely reflect a decline in open-water calcite formation, a delayed response to P reduction measures that began in the late 1970s (*Barbiero, Tuchman & Millard, 2006*). Later, Ca uptake by invasive dreissenid mussels, which were first detected in Lake Erie in 1988, probably further contributed to the decline in Ca concentrations in these deepwater sediment core locations.

## Sediment contamination guidelines

No literature could be found that suggested there is environmental concern regarding sediment metals in the deepest, pelagic regions of the Great Lakes. However, exceedances of sediment contamination guidelines were common, especially in Lake Ontario. Sometimes these exceedances occurred in pre-industrial sediments, so it is worth considering the natural or "baseline" context acquired from long-term sediment records to confirm whether contamination is a result of human activities and to determine appropriate

remedial measures. Most of the guideline exceedances are from long-buried sediments, so environmental threat may be minimal assuming these deep sediments are never dredged. With all of these caveats considered, our examination of surface sediment metals suggests Pb in all lakes, Cu in Lakes Ontario and Superior, and Cr in Lakes Erie and Ontario may be problematic in that these high concentrations of contaminant metals are available to benthic biota and so may contribute to food web bioaccumulation. As and Mn values in recent sediments are also above the extreme effect level in all lakes but this may be due to redox-induced mobilization as discussed above.

## CONCLUSIONS

Using paleolimnological methods, we were able to construct a history of metals pollution throughout the Great Lakes basin. Certain trends are observed basin-wide, such as for atmospheric pollutants including Pb, which follows trends related to the use and ban of leaded gasoline. Other trends are lake-specific, such as high levels of Na in Lake Superior due to road salt applications and strong evidence of substantial whiting events in Lake Ontario. The Great Lakes are sensitive to environmental changes such as pollution by metals, and it is clear that, while there has been remedial success, results from the uppermost intervals of cores indicate ongoing enrichment of contaminants. The impact of certain management practices, such as the banning of leaded gasoline and improvement of municipal sewage, is visible in the cores and demonstrates the efficacy of these practices. Certain analytes, especially Na, are still increasing in sedimentary records indicating they may require further management. For future studies, mineralogical contents of the cores should be considered so that elements can be more accurately characterized to natural sources. Further, extension of coring efforts to nearshore areas would allow for better identification of the sources of remaining contaminants of concern.

## ACKNOWLEDGEMENTS

This document has not been subjected to the EPA's required peer and policy review and therefore does not necessarily reflect the view of the Agency, and no official endorsement should be inferred. We thank Kathleen Kennedy, Amy Kireta, Robert Sterner, Craig Stow and the Research Vessel Lake Guardian and Blue Heron field crews for their help collecting core samples. Sediment dating was supported by Daniel Engstrom and personnel at the St. Croix Watershed Research Station. James Latimer and two anonymous reviewers provided valuable comments on this manuscript.

### Funding

This research was supported by: (A) a grant to Euan Reavie from the U.S. Environmental Protection Agency under Cooperative Agreement GL-00E23101-2; and (B) State Special funding from Minnesota to the Natural Resources Research Institute. There was no

additional external funding received for this study. The funders had no role in study design, data collection and analysis, decision to publish, or preparation of the manuscript.

### Grant Disclosures

The following grant information was disclosed by the authors:
U.S. Environmental Protection Agency: GL-00E23101-2.
Natural Resources Research Institute.

### Competing Interests

The authors declare there are no competing interests.

### Author Contributions

- Malachi Nicholas Granmo conceived and designed the experiments, performed the experiments, analyzed the data, prepared figures and/or tables, authored or reviewed drafts of the paper, and approved the final draft.
- Euan D. Reavie conceived and designed the experiments, performed the experiments, analyzed the data, authored or reviewed drafts of the paper, and approved the final draft.
- Sara P. Post analyzed the data, prepared figures and/or tables, authored or reviewed drafts of the paper, and approved the final draft.
- Lawrence M. Zanko analyzed the data, authored or reviewed drafts of the paper, and approved the final draft.

### Data Availability

The raw data and R scripts used to generate Table 1 and Figures 2–10 are available in the Supplemental Files.

### Supplemental Information

Supplemental information for this article can be found online at http://dx.doi.org/10.7717/peerj.9034#supplemental-information.

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
