# Peer review of "Anthropocene geochemistry of metals in sediment cores from the Laurentian Great Lakes"

_PeerJ, doi:10.7717/peerj.9034_

## Round 0.1 · original submission · Major Revisions

I believe that your article makes an important contribution to the understanding of metal and other elemental geochemistry in the Great Lakes. However, I believe that reviewer 2 did identify some deficiencies that need to be address before the article is accepted.

Reviewer 1 ·

Basic reporting

Clear, professional English is used throughout.

Experimental design

The major strength of this study is the extensive literature review.

Validity of the findings

no comment

Additional comments

The manuscript titled “Anthropocene geochemistry of metals in sediment cores from the Laurentian Great Lakes” by Meagan Aliff et al. reviews a set of sediment geochemical records from the Great Lakes to estimate anthropogenic and natural changes in the metal inputs to the Lakes over the last ~200 years. Using cluster analysis, the co-authors found five distinct clusters of metals related to distinct natural (e.g., whiting events) and anthropogenic processes (e.g., mining). In particular, the co-authors noted that certain anthropogenic contaminates, such as Pb, were observed to increase across all the basins, and was associated with the rise in the use of tetra-ethyl leaded gasoline. The co-authors’ findings are relevant to ongoing remediation efforts in the Lakes, and the extensive review of the available literature itself is a valuable contribution. I do have some suggests.

Lines 38-40: While the Native American influence on the Great Lakes was undoubtably smaller than Euro-American impacts, increasing concentrations of metals to the Great Lakes certainly did not begin with “…extensive European settlement around 1850…” Findings by Pompeani et al. (2015) indicate that extensive pre-historic copper mining on Isle Royale made a substantial, measurable impact to the delivery of Pb and Cu to the sediments in Lake Superior. I would assume that there are likely other pre-historic copper mining impacts to Lake Superior in the Michigan Copper Districts as well. I suggest that the co-authors choose their wording carefully to include the potential for Native American impacts, while maintaining that the Euro-American environmental impact after 1850 was likely larger.

Line 51: Pollutants are mobilized and deposited in lakes via the atmosphere, in addition to rivers. This is particularly important for Pb due to the burning of tetra-ethyl leaded gasoline and smelting. I would include atmospheric transport processes in this sentence as well.

Lines 71-75: Historical Pb concentrations, which likely reflect leaded gasoline combustion, have also been studied in Lake Superior by Pompeani et al. (2015). This citation should be included in this section as well.

Lines 187-190: It is not clear why the Barbieri citation is used here. This EF equation was developed much earlier, which I think would be more relevant to cite. For example, see Weiss et al. (1999) and citations therein. Also, why is the subscript labeled “soil” in the equation? This study analyzes lake sediment, not soil. This needs to be changed or clarified in the text.

Line 240: I believe P2O5 is technically phosphorous pentoxide, not phosphate (which is PO4-3).

Lines 306-326: It is not clear why the study by Pompeani et al. (2015) is not included in this section, especially because they present Pb, K, Fe, and Cu concentrations from lake sediment cores in Lake Superior for >200 years. This citation should be included in this section, relevant statistical analyses, and figures.

Line 531: To expand the distribution of sample cores, I would include the sediment core studied by Pompeani et al. (2015).



Pompeani, D. P., M. B. Abbott, D. J. Bain, S. DePasqual, and M. S. Finkenbinder
2015 Copper mining on Isle Royale 6500-5400 years ago identified using sediment geochemistry from McCargoe Cove, Lake Superior. The Holocene 25(2):253-262.

Weiss, D.;, W.; Shotyk, P. G.; Appleby, J.D.; Kramers, and A.K. Cheburkin
1999 Atmospheric Pb deposition since the industrial revolution recorded by five Swiss peat profiles: enrichment factors, fluxes, isotopic composition and sources. Environmental Science and Technology 33:1340-1352.

Reviewer 2 ·

Basic reporting

This paper reports on the analysis of 50 metals/metalloids and 10 metal oxides in 11 sediment cores collected from all five Great Lakes. It is clearly written and structured. The introduction provides reasonable context although studies of metal(loids) in dated sediment cores from outside of the Great Lakes are not cited. Tables and figures are informative although Figure 5-10 are difficult to read due to small size of each panel and 11 profiles in each one

Experimental design

The research questions are well defined in the introduction. The work appears to be performed to a high technical standard but analytical methods are described very briefly and no QA/QC information is given. There is a reference to Reavie et al 2020 in review (line 475) which may address this but it was not available as part of the Supplementary Info.
The manuscript also suffers from the lack of information on sediment core dating. It is unusual to not have it available within the same manuscript given the importance in linking to evidence for historical emissions of pollutants. Apparently it is in another manuscript but unavailable for review. There are some odd comments eg that 137Cs was determined only on one core (line 112) which is surprising given that it is usually determined alongside 210Pb and 214Pb by gamma spectrometry.

Validity of the findings

The study has many unique aspects starting with the large suite of trace metals, the analysis of the oxides in the same samples, and coverage of all lakes. Previous work on metals has generally not covered all Great Lakes in the same study (unlike the case for organic contaminants). There are also some novel results, for example, the observations on increasing sodium in the sediment and the relationship to increasing use of road salt is quite remarkable. It is also unusual to see measurements of the more soluble salts in sediment cores. In fact, on a quick search using Google Scholar I was unable to find anything specifically on that.
Given the large amount of data the approach of using cluster analysis and PCA to define groups of elements, and then discussing by group for each lake, is practical and appropriate. On the other hand the grouping means that a lot of elements are not discussed in any detail. An example is zinc which shows large EFs (eg 4X in Lake Ontario), is combined with Cd, Sn, Sb, and Pb. Zn is the 4th most widely used metal globally and has important sources due to increased use for galvanizing steel etc, while Pb emissions have declined as the authors have noted. So the two differ greatly in the source function and yet are discussed together. Thus there is the potential for a series of papers where there is more emphasis on sources of individual elements.
While the historical concentration profiles are very interesting, their interpretation is confounded by changes in sedimentation rates (provided in metals3.csv – as “accum rate” – although with no units). In some cores the sedimentation rate has doubled (eg Erie Central) while in others (eg Ontario East) it has halved over the past 50 years. It would have been far better to report trends as fluxes to discuss historical trends and also ratios such as EF and IGeo in order to avoid obvious issues of dilution due to sedimentation of allochthonous or autochthonous particles.

Additional comments

In summary I think the authors should reconsider the way the manuscript is organized such as splitting it into multiple articles focusing on specific groups of elements while including dating information. Or perhaps they should wait until the Reavie et al article is published so that reviewers have more information on dating and methods.

Specific comments
line 15. This is not the case for organic contaminants - there have been numerous basis wide studies with dated sediment cores
line 112. Unfortunate that the dating information is not available because there can be difficulties interpreting the 210Pb data. Surprising that 137Cs would not be available given that it is determined with 210Pb by gamma spectrometry
Line 112 – Supplemental data. Are the results for total metals in the SI (metals3.csv) eg for Al and Ca, after subtracting the oxides? The digestion and ICP-MS give total?
line 125. Unclear how many elements were determined with this method. Authors should comment on why they did not use a total extraction (aqua-regia - HF) method. Also QA information such as use of certified reference materials is not provided. References should be given for the methods and rationale for their selection.
line 140. It's unclear how the oxides were actually determined or which metalswere done with ICP MS vs ICP-OES. Also what about reference materials for the oxides?
line 143. This sentence should be deleted ie by simply stating that Hg was not analysed. Given the different emission sources and controls, which are fairly recent in the case of Hg, it is not appropriate to cite make a comparison with Pb based on a 27 year old paper
line 293. Why were Li and Ti not measured in all cores?
line 296. Should clarify that this is not a core from lakes on Isle Royale but simply a mid-lake core.
line 322 and elsewhere. Suggest using µg/g throughout since ppm can be misinterpreted
line 475. This could be corrected by using fluxes instead of concentrations
line 543. Duluth is actually lower population than Thunder Bay, but the metropolitan area including Superior WI has greater population than the Thunder Bay region
line 581. Forest fires could be very important for Lake Superior as well. This information is available for US and for Ontario
Table 1. Why were only some results Bonferroni corrected?
Figure 2. Why not use hectares or square km to be consistent with descriptions of farmland globally
Figure 5-10 are difficult to read due to small size of each panel and 11 profiles in each one

---

## Round 0.2 · accepted · Accept

After a careful examination of the authors' responses to reviewers comments, I am satisfied that they have adequately addressed the shortcomings of the original submission.